# COVID-19 and pregnancy: An umbrella review of clinical presentation, vertical transmission, and maternal and perinatal outcomes

Agustín Ciapponi[1]*, Ariel Bardach[1], Daniel Comandé[1], Mabel Berrueta[1], Fernando J. Argento[1], Federico Rodriguez Cairoli[1], Natalia Zamora[1], Victoria Santa María[1], Xu Xiong[2], Sabra Zaraa[3], Agustina Mazzoni[1], Pierre Buekens[2]

1 Instituto de Efectividad Clínica y Sanitaria (IECS-CONICET), Buenos Aires, Argentina, 2 School of Public Health and Tropical Medicine, Tulane University, New Orleans, LA, United States of America, 3 School of Pharmacy, University of Washington, Seattle, WA, United States of America

* aciapponi@iecs.org.ar

## Abstract

### Background

We conducted an overview of systematic reviews (SRs) summarizing the best evidence regarding the effect of COVID-19 on maternal and child health following Cochrane methods and PRISMA statement for reporting (PROSPERO-CRD42020208783).

### Methods

We searched literature databases and COVID-19 research websites from January to October 2020. We selected relevant SRs reporting adequate search strategy, data synthesis, risk of bias assessment, and/or individual description of included studies describing COVID-19 and pregnancy outcomes. Pair of reviewers independently selected studies through COVIDENCE web-software, performed the data extraction, and assessed its quality through the AMSTAR-2 tool. Discrepancies were resolved by consensus. Each SR's results were synthesized and for the most recent, relevant, comprehensive, and with the highest quality, by predefined criteria, we presented GRADE evidence tables.

### Results

We included 66 SRs of observational studies out of 608 references retrieved and most (61/66) had "critically low" overall quality. We found a relatively low degree of primary study overlap across SRs. The most frequent COVID-19 clinical findings during pregnancy were fever (28–100%), mild respiratory symptoms (20–79%), raised C-reactive protein (28–96%), lymphopenia (34–80%), and pneumonia signs in diagnostic imaging (7–99%). The most frequent maternal outcomes were C-section (23–96%) and preterm delivery (14–64%). Most of their babies were asymptomatic (16–93%) or presented fever (0–50%), low birth weight (5–43%) or preterm delivery (2–69%). The odds ratio (OR) of receiving invasive ventilation for COVID-19 versus non-COVID-19 pregnant women was 1.88 (95% Confidence Interval [CI] 1.36–2.60) and the OR that their babies were admitted to neonatal

**Data Availability Statement:** All relevant data are within the manuscript and its Supporting Information files.

**Funding:** This work was supported, in whole by the Bill & Melinda Gates Foundation [INV008443]. Under the grant conditions of the Foundation, a Creative Commons Attribution 4.0 Generic License has already been assigned to the Author Accepted Manuscript version that might arise from this submission. The sponsors had no role in conducting the present study.

**Competing interests:** The authors have declared that no competing interests exist.

intensive care unit was 3.13 (95%CI 2.05–4.78). The risk of congenital transmission or via breast milk was estimated to be low, but close contacts may carry risks.

## Conclusion

This comprehensive overview supports that pregnant women with COVID-19 may be at increased risk of adverse pregnancy and birth outcomes and low risk of congenital transmission.

## Introduction

Women undergoing pregnancy, and those at the time of childbirth and puerperium constitute potentially vulnerable populations for COVID-19. Although our understanding of this disease is growing every day, many answers are still needed about the diagnostics and the clinical management methods in these groups, the impact of the disease in pregnant women and newborns, and the potential of mother-to-child transmission.

Although some living guidelines on COVID-19 target the pregnancy population, several clinical questions regarding pregnancy and childbirth remain unanswered [1]. The rate of COVID-19 in pregnant and recently pregnant women attending or admitted to hospital for any reason was around 10%. Pregnancy, in general, does not significantly increase the risk of being infected by SARS-CoV-2 [2].

The World Health Organization (WHO) stated that pregnant women or recently pregnant women who are older, overweight, and have pre-existing medical conditions such as hypertension and diabetes seem to have an increased risk of developing severe COVID-19 [3]. In general, there is a consensus that breastfeeding should be promoted due to its mutual benefits. However, it is not well known whether the virus can be transmitted through breastmilk [4].

Systematic reviews (SRs) constitute an organized effort to collect and comprehensively synthesize the best available evidence on a given topic. Through this panoramic review of SRs, we aimed to answer a series of clinical questions about COVID-19 and pregnancy by summarizing the body of evidence and highlighting the best reviews in completeness and methodological quality.

## Objectives

To summarize the clinical presentation, vertical transmission, and maternal and perinatal outcomes in pregnant women with COVID-19 and their neonates.

## Methods

We performed an overview of SRs or umbrella review (PROSPERO Registration number CRD42020208783) following Cochrane methods [5] and the Preferred Reporting Items for systematic Reviews and Meta-Analyses (PRISMA) statement [6] and a specific guideline for overviews [7] (**S1 File**) for reporting.

## Eligibility criteria

We included SRs that met the Database of Abstracts of Reviews of Effects (DARE) criteria [8]: 1) reported eligibility criteria, 2) adequate search, 3) data synthesis, 4) risk of bias assessment and/or 5) individual description of included studies.

To be included, SRs had to meet at least four of these criteria, the first three of which were mandatory. The exposures of interest were defined as diagnosis of SARS-CoV-2 infection, SARS-CoV-2 risk factors, diagnostic tests, or treatments. Pregnant women without interventions or exposures under study, including active or inactive comparators, usual care, or placebo, were defined as comparison groups. Any pregnancy or neonatal outcomes, including clinical presentation, laboratory, and radiological findings, were included (**S2 File**).

## Search strategy

From January to October 2020 an experienced librarian searched the Cochrane Library, MED-LINE, EMBASE, Latin American and Caribbean Health Sciences Literature (LILACS), Science Citation Index Expanded (SCI-EXPANDED), China Network Knowledge Information (CNKI), WHO Database of publications on SARS-CoV-2, EPPI-Centre map of the current evidence on COVID-19, guidelines published by national and international professional societies (e.g., ACOG, RCOG, FIGO), pre-print servers (ArXiv, BiorXiv, medRxiv, search.bioPreprint), and COVID-19 research websites (PregCOV-19LSR, Maternal and Child Health, Nutrition: John Hopkins Centre for Humanitarian health, the LOVE database) We also searched the reference lists of included SRs. No language or publication status restrictions were applied (The whole search strategy is presented in the **S3 File**).

## Study selection data extraction and quality appraisal

Pairs of reviewers independently screened titles and abstracts. We retrieved all potentially relevant full-text study reports/publications, and two reviewers independently evaluated the full-texts, recording the reasons for exclusion of the ineligible studies. Disagreements were resolved through discussion of the review team. This process was performed using the web-based software COVIDENCE [9].

Pairs of reviewers independently performed the data extraction through an online extraction form previously piloted in five studies. We recorded publication date, number of included studies, number of included participants, quality items, and the components of our research questions (population, exposition, comparisons, and outcomes). Discrepancies were resolved by consensus.

Pairs of reviewers independently assessed the quality of SRs through the AMSTAR-2 tool [10]. The instrument has 16 items. It is not intended to generate an overall score but provides a categorical rating based on critical domains: protocol register, adequacy of the literature search, justification for excluding individual studies, risk of bias from individual studies being included, appropriateness of meta-analytical methods, consideration of risk of bias when interpreting the results, assessment of publication bias. The overall quality or confidence in the results of the review can be rated as "high" (no or one non-critical weakness), "moderate" (more than one non-critical weakness), "low" (one critical flaw with or without non-critical weaknesses), and "critically low" (more than one critical flaw with or without non-critical weaknesses). Discrepancies were resolved by consensus. We did not assess the quality of the included primary studies in the SRs nor the quality of reporting of each SR.

## Synthesis of results

An analysis of the overlap of the primary studies included by each systematic review was performed. Only primary articles with DOI numbers were included for this analysis. We presented all the outcome measures reported in the SRs with proportions, relative risks, odds ratios, risk difference, and/or 'number needed to treat' mean differences, standardized mean differences, with 95% confidence intervals.

The purpose of our study is to present and describe the current body of SRs evidence on COVID-19 in maternal and neonatal health. Therefore, we synthesized the results of all relevant SRs, regardless of topic overlap, considering that re-extracting and re-analyzing outcome data from non-overlapping studies was unfeasible and outside the scope of this overview. Additionally, for this scoping synthesis, we selected the best SR that answers a specific question according to pre-defined prioritizing criteria: most relevant, most comprehensive, most recent, and highest quality determined by AMSTAR-2 [10]. For these prioritized reviews, pairs of reviewers independently assessed the risk of bias of the priori SR using the tool Risk of Bias in Systematic Review (ROBIS) [11] and the GRADE approach evaluating the certainty of evidence of each outcome [12, 13]. We did not assess the quality of the included primary studies in the SRs nor the quality of reporting of each SR.

We presented summaries of the findings in a format suitable for decision-makers, previously validated during the SUPPORT project [14], focusing on low- and middle-income countries (LMICs) for selected research questions.

Pre-specified subgroups were designated by sampling frame (universal, symptom-based, or risk-based testing), timing of suspicion/diagnosis (pregnancy or postnatal period), trimester of suspicion/diagnosis (first, second or third), country income-level (high or low- and middle-income country), and maternal risk status (low or high).

## Results

The cumulative search retrieval was 608 records, 126 potentially eligible reviews were assessed by full-text and 66 were included for clinical presentation in pregnant women (n = 39), maternal outcomes (n = 44), clinical presentation in neonates (n = 28), neonatal outcomes (n = 41) and vertical transmission (n = 46) (**Fig 1**).

**Table 1** presents the included studies and **S4 File** the list of excluded studies with their exclusion reasons. All the included SRs were conducted during 2020 all over the world. Although all of them reported a qualitative summary, 18 also included a quantitative summary [2, 15–31].

Among the primary studies included in these reviews, the main study designs were case reports, case series, and other observational studies with or without a comparison group.

The number of included SARS-CoV-2 positive pregnant women was highly heterogeneous across reviews. While the Li review [32] and the Mullins review [33] included only 19 of them, the Allotey review [2] included more than ten thousand.

**Fig 2** shows the percentual degree of overlap of included systematic reviews' primary studies, which in general was low (In **S5 File** we listed all primary studies with DOI included by each review and in **S6 File** the degree of overlap in absolute numbers).

Concerning the overall quality, based on AMSTAR-2 (see **Table 1**), most SRs were classified as "critically low" (n = 61), four as "low" [2, 34–36], and only one as "moderate" [37]. For the prioritized systematic reviews, we also used the ROBIS tool (one was classified as "low risk of bias" [2], one as "unclear risk of bias" [38] and the other two as "high risk of bias" [37, 39]) (**S7 File**, by each domain). The mean ± standard deviation of non-negative classifications was 9.21 ± 2.51. The most common weaknesses (>50% of SRs with unmet domain) were: not reporting the funding for the studies included in the review (n = 65), not providing a list of excluded studies and justifying the exclusions (n = 61), not accounting for risk of bias (RoB) in individual studies when interpreting or discussing the results of the review (n = 50), not providing a satisfactory discussion of any heterogeneity observed (n = 50), not providing a protocol (n = 47) and not using satisfactory techniques for assessing the RoB in individual studies included in the review (n = 39).

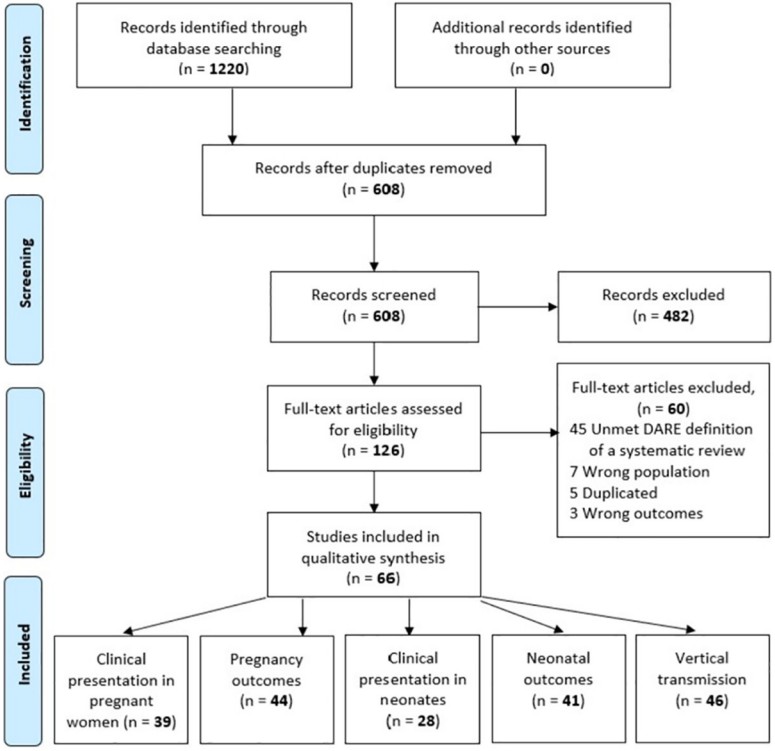

**Fig 1. Study flow diagram.**

Below the main findings are described for the five available bodies of evidence identified in our overview: clinical presentation in pregnant women, pregnancy outcomes, clinical presentation in neonates, neonatal outcomes, and vertical transmission.

There was a high level of heterogeneity of the reported values for each outcome across the included SRs. **Table 2** shows these value ranges (The **S8 File** shows the total number of newborns and pregnant women included and the numerical data of each outcome at the review level).

The main findings are described briefly below.

## Clinical presentation in pregnant women

Thirty-nine reviews [2, 15, 16, 18, 19, 21–23, 25, 28–37, 39–58] showed information regarding clinical manifestations during pregnancy. Fever and mild respiratory symptoms were the most frequently reported symptoms and raised C-reactive protein level, lymphopenia, raised white cell count and raised procalcitonin level were the most frequent laboratory findings. Signs of pneumonia on X-rays or computed tomography (CT) were also frequently reported (**Table 2, Tables 1, 2 in S8 File**).

The Allotey review [2] was chosen as the best review for reporting clinical presentation in pregnant women diagnosed with COVID-19 according to predefined criteria described in the methods section (**Table 3** and **S9 File**). It reported that fever (40%), cough (39%, involving 28 studies and 8317 pregnant women), and dyspnea (19%) were the most common symptoms. It also reported that raised C reactive protein levels (49%) and lymphopenia (35%) were the most common laboratory findings. Regarding findings on X-rays or CT, ground glass appearance had a prevalence of 69%, and any other abnormalities on CT had a prevalence of 65%. Finally,

**Table 1. Main characteristics, research questions and findings of included systematic reviews.**

| Authors (2020) | Questions (Q) | Search Date[#] | (Q) N studies | (Q) N women | (Q) N neonates | AMSTAR-2 (1 to 16)[*] | Overall confidence | Main findings |
|---|---|---|---|---|---|---|---|---|
| AbdelMassih [15] | 4 | 8/1/20 | 66 | 1787 | 1787 | 6 | Critically low | Only 2.8% of mother-infant pairs were tested positive, and this finding is identical to percentages reported in former coronaviridae outbreaks. In contrast, 20% manifested with intrauterine hypoxia alongside placental abnormalities suggestive of heavy placental vasoocclusive involvement. |
| Abdollahpour [59] | 2, 3, 4 | 3/25/20 | 2 (29), 4 (4) | NA | NA | 10 | Critically low | The clinical features of these patients with COVID-19 infection during pregnancy were parallel to those of non-pregnant adults with COVID-19 infection. The main symptoms of pregnant women with COVID-19 were fever and cough. Leukocytosis (41%) and elevated neutrophil ratio (83%) were unusually noted. Symptoms are mild to moderate in pregnancy. No trustworthy evidence is available yet to support the possibility of vertical transmission of COVID-19 infection from the mother–baby. Mother-to-child transmission of respiratory viruses mostly happens via the birth canal and during breastfeeding or close contact. |
| Akhtar [40] | 2, 3 | 5/22/20 | 22 | 156 | 108 | 12 | Critically low | Most of the mothers received nasal oxygen therapy; many received antiviral and antibiotic medications. Maternal clinical manifestations reported were fever (53%), cough (32%), fatigue/malaise (13%), myalgia (11%), sore throat (5%), and shortness of breath (8%). A marked lymphopenia was also noted in many patients with COVID-19. The most common maternal/fetal complications included intrauterine/fetal distress (14%) and premature rupture of membranes (PROM) (8%). The neonatal clinical manifestations of COVID-19 commonly included shortness of breath (6%), gastrointestinal symptoms (4%), and fever (3%). |
| Allotey [2] | 2, 3 | 6/26/20 | 2 (40), 3 (23–35) | 2 (13018), 3 (6279–95247) | | 14 | Low | The spontaneous preterm birth rate was 6% (95% CI 3% to 9%) in women with COVID-19. The odds of any preterm birth (3.01, 1.16 to 7.85) were high in pregnant women with COVID-19 compared with those without the disease. A quarter of all neonates born to mothers with COVID-19 were admitted to the neonatal unit (25%) and were at increased risk of admission (OR 3.13, 2.05 to 4.78) than those born to mothers without COVID-19. Increased maternal age (1.78, 1.25 to 2.55), high body mass index (2.38, 1.67 to 3.39), chronic hypertension (2.0, 1.14 to 3.48), and pre-existing diabetes (2.51, 1.31 to 4.80) were associated with severe COVID-19 in pregnancy. |
| Arabi [16] | 2, 3 | 3/20/20 | 7 | 50 | | 8 | Critically low | Seven studies involving 50 participants with Positive test of COVID-19 were enrolled. Same clinical characteristics in pregnant women as in non-pregnant adults were observed, with cough and fever as prominent symptoms. No vertical transmission was seen. |

(*Continued*)

**Table 1.** (Continued)

| Authors (2020) | Questions (Q) | Search Date[#] | (Q) N studies | (Q) N women | (Q) N neonates | AMSTAR-2 (1 to 16)* | Overall confidence | Main findings |
|---|---|---|---|---|---|---|---|---|
| Ashraf [41] | 2, 3, 4 | 4/14/20 | 21 | 90 | 2 (86), 4 (90) | 9 | Critically low | The most common symptoms included fever, cough, and dyspnea. The main laboratory findings included leukocytosis, lymphopenia, thrombocytopenia, and elevated C-reactive protein. The most commonly reported complications were preterm labor and fetal distress. Three mothers were admitted to ICU and required mechanical ventilation; among them, one died, and one was on extracorporeal membrane oxygenation. 82/86 neonates were negative in RT-PCR, while four were positive. Out of 92 neonates, one died, and one was born dead. |
| Banaei [60] | 2, 3, 4 | 4/10/20 | 13 | 123 | 35 | 11 | Critically low | The result of five neonates was positive for SARS-CoV-2. |
| Bwire [73] | 4 | 5/18/20 | 33 | NA | 205 | 12 | Critically low | The current evidence revealed a low possibility of vertical transmission of COVID-19, and antibodies against SARS-CoV-2 were detected among vertically exposed but negative infants. |
| Caparros Gonzalez [66] | 3, 4 | 3/1/20 | 5 | 33 | NA | 6 | Critically low | All empirical studies in this review reported an absence of vertical transmission of the coronavirus from the pregnant mother to the developing fetus |
| Centeno-Tablante [38] | 4 | 5/15/20 | 37 | 77 | 77 | 14 | Critically low | 19 of 77 children (25%) were confirmed COVID-19 cases based on RT-PCR assays, including 14 neonates and five older infants. Nine of the 68 analyzed breast milk samples from mothers with COVID-19 were positive for SARS-CoV-2 RNA; of the exposed infants, four were positive, and two were negative for COVID-19. |
| Chi [61] | 3, | 3/31/20 | 14 | 107 | 3 (105), 4 (91) | 10 | Critically low | 43 mothers developed perinatal complications, including preeclampsia, placenta previa, placenta abruptio, fetal distress, premature rupture of membranes, and uterine scarring. |
| de Sousa [42] | 2, 3, 4 | 5/26/20 | 49 | 755 | 3 (598), 4 (493) | 8 | Critically low | No evidence of vertical transmission based on what has been assessed so far |
| Della Gatta [43] | 2, 3 | 3/16/20 | 6 | 51 | | 11 | Critically low | Six studies that involved 51 pregnant women were eligible for the systematic review. At the time of the report, three pregnancies were ongoing; of the remaining 48 pregnant women, 46 gave birth by cesarean delivery, and 2 gave birth vaginally; in this study, one stillbirth and one neonatal death were reported. |
| Deniz [62] | 2 4 | 6/1/20 | 50 | 606 | 606 | 7 | Critically low | Twenty neonates had evidence of SARS CoV-2 infection (positive RT-PCR results or elevated level of SARS CoV-2 antibodies in serum samples) |
| Dhir [63] | 3, 4 | 6/9/20 | 45 | 1992 | 3 (1125), 4 (1005) | 6 | Critically low | A total of 1141 neonates were born of which, 281 (25%) were preterm (<37weeks). SARS-CoV-2 testing was positive for 39/1005 neonates (3.9%), 16/43 mother-baby dyads (37.2%) were preterm (<37weeks), 9 (21%) were low birth weight (<2500 g), and 27 (62.8%) were born by cesarean section (Table 2). All 43 were tested for SARS-CoV-2 infections using nasopharyngeal or oropharyngeal specimen, and 19 neonates (44.2%) have positive RT-PCR for SARS-CoV-2.58 live-born SARS-CoV-2 cases, 4 (7%) were congenital in origin (2 confirmed, one probable, and one not sure), 41 were acquired in the postpartum period, and the remaining 13 neonates could not be classified due to non-available |

*(Continued)*

**Table 1.** (Continued)

| Authors (2020) | Questions (Q) | Search Date# | (Q) N studies | (Q) N women | (Q) N neonates | AMSTAR-2 (1 to 16)* | Overall confidence | Main findings |
|---|---|---|---|---|---|---|---|---|
| Di Mascio [17] | 2, 3, 4 | 3/13/20 | 2 (19), 3 (16), 4 (6) | 2 (79), 3 (58) | 2 (60), 4 (42) | 10 | Critically low | An overt diagnosis of pneumonia was made in 9,1.8%nd the most common symptoms were fever (82.6%), cough (57.1%), and dyspnea (27.0%). The pooled proportion of cesarean delivery was 83.9% (95%CI 73.8–91.9), perinatal death was 11.1% 84.8–19.6), including three stillbirths and two neonatal deaths. A total of 34.2% (20.3–49.5) of fetuses suffered from fetal distress, and 57.2% (3.6–99.8) of newborns were admitted to the NICU. None of the newborns showed signs of vertical transmission |
| Diriba [18] | 2, 3, 4 | 4/30/20 | 25 | 1271 | 1271 | 12 | Critically low | None of the studies reported transmission of SARS-CoV2 from the mother to the fetus in utero during the follow-up period |
| Duran [67] | 2, 3, 4 | 4/17/20 | 20 | NA | 222 | 7 | Critically low | Out of the 222 newborns, 13 were reported as positive for SARS-CoV-2. Five of the 20 studies reported data on umbilical cord blood, placenta, and/or amniotic fluid, all with no positive results. |
| Figueiro-Filho [39] | 2, 3 | 7/1/20 | 8 | Depends on each outcome | Depends on each outcome | 9 | Critically low | We suggest that pregnant women are not more affected by the respiratory complications of COVID-19 when compared to the outcomes described in the general population. |
| Furlan [44] | 2, 3, 4 | 3/1/20 | 2 (27), 4 (22) | 399 | 80 | 10 | Critically low | The most common symptoms reported by pregnant women with COVID-19 were fever and cough Positive SARS-CoV2 test in neonates: 4/80 |
| Gajbhiye [45] | 2, 3, 4 | 5/3/20 | 2 (50), 4 (33) | 2 (441), 4 (327) | 2 (291), 4 (395) | 8 | Critically low | There are nine maternal deaths reported. In pregnant women, the most common symptoms were fever (56%), cough (43%), myalgia (19%), dyspnea (18%), and diarrhea (6%). Pregnancy complications included delivery by cesarean section (80%), preterm labor (26%), 44 fetal distress (8%) and premature rupture of membranes (9%). Amongst the neonates of COVID-19 mothers, 48 preterm birth (25%), respiratory distress syndrome (8%), pneumonia (8%) were reported. There 49 were four neonatal deaths reported. Pneumonia was diagnosed by CT scan imaging in 96% of COVID-19 pregnant women. Vertical transmission rate of SARS-CoV-2 is estimated to be 8%. |
| Gao [19] | 2, 3 | 4/16/20 | 14 | 236 | | 9 | Critically low | Positive CT findings (71%; 95%CI 0.49–0.93), fever (51%; 0.35–0.67), lymphopenia (49%; 0.29–0.70), cough (31%; 0.23–0.39), fetal distress (29%; 0.08–0.49). Compared with non-pregnant patients, pregnant women with COVID-19 had significantly lower incidences of fever (pregnant women, 51%; non-pregnant patients, 91%; P < 0.00001) and cough (pregnant women, 31%; non-pregnant patients, 67%; P < 0.0001). fetal distress (29%; 0.08–0.49), preterm labor (23%; 0.14–0.32), and severe case or death (12%; 0.03–0.20). |
| Goh [20] | 4 | 3/23/20 | 17 | 402 | 405 | 3 | Critically low | The average pooled incidence of vertical transmission was 16 per 1000 newborns (95%CI 3.40 to 73.11) |
| Gordon [68] | 4 | 5/12/20 | 8 | NA | 40 | 13 | Critically low | Of the ten reported cases, only three are likely to be vertically transmitted, while seven occurred in the post perinatal period and are likely to have been postnatally acquired. All neonates had a mild course, recovered, and were negative on re-testing. |

(Continued)

**Table 1.** (Continued)

| Authors (2020) | Questions (Q) | Search Date# | (Q) N studies | (Q) N women | (Q) N neonates | AMSTAR-2 (1 to 16)* | Overall confidence | Main findings |
|---|---|---|---|---|---|---|---|---|
| Han [21] | 2, 3 | 6/10/20 | 36 | Depends on each outcome | Depends on each outcome | 7 | Critically low | Pregnant patients with COVID-19 most commonly presented with fever, cough, shortness of breath and dyspnea, most of which possessed imaging manifestations. The risk of premature delivery was higher, leading to a high risk of NICU admission and low neonatal birthweight. Vertical transmission was found to be unlikely. |
| Hasan [74] | 2, 3 | 3/31/20 | 29 | NA | NA | 11 | Critically low | Evidence of higher perinatal complications puts pregnant women in a further vulnerable condition. Cautiousness is imperative during the clinical management of pregnant women with COVID-19. |
| Hessami [75] | 2, 3 | 7/20/20 | 10 | 37 | 12 | 7 | Critically low | 37 maternal and 12 perinatal mortality cases. All maternal deaths were seen in women with previous comorbidities, of which the most common were obesity, diabetes, asthma, and advanced maternal age. Acute respiratory distress syndrome and severity of pneumonia were considered as the leading causes of all maternal mortalities. Fetal and neonatal mortalities were suggested to be a result of the severity of maternal infection or prematurity, respectively. There was no evidence of vertical transmission. |
| Huntley [34] | 4 | 4/29/20 | 10 | 310 | 310 | 14 | Low | There were no cases of vertical transmission among 310 deliveries for which reverse-transcription polymerase chain reaction data were made available. |
| Juan [37] | 4 | 4/20/20 | 18 | 174 | 174 | 14 | Moderate | Throat swab test in neonates: 4/174 |
| Kasraeian [22] | 2, 3 | 3/18/20 | 9 | 87 | 86 | 7 | Critically low | No evidence of vertical transmission has been suggested at least in late pregnancy. No hazards have been detected for fetuses or neonates. Although pregnant women are at an immunosuppressive state due to the physiological changes during pregnancy, most patients suffered from mild or moderate COVID-19 pneumonia with no pregnancy loss. |
| Khalil [23] | 4 | 6/8/20 | 2 (17), 4 (69) | 2 (2576), 4 (290) | | 13 | Critically low | The most reported clinical symptoms were fever (63.3%), cough (71.4%), and dyspnea (34.4%). The commonest laboratory abnormalities were raised CRP or procalcitonin (54.0%), lymphopenia (34.2%), and elevated transaminases (16.0%). Analysis of conception products that may be associated with vertical transmission was reported in a minority of cases (placenta: 10.7%, amniotic fluid: 5.5%, cord blood: 6.2%). Maternal bodily fluid PCR positivity was rare (vaginal swab: 0%, stool: 12.5%, breast milk: 6.7%). |
| Khan [46] | 2 | 3/25/20 | 9 | 2 (101), 3 (60) | 56 | 9 | Critically low | Fever (66.7%), cough (39.4%), fatigue (15.2%), and breathing difficulties (14.1%) were common. Of all deliveries that occurred, 83.9% had gone through the C-section, and around 30.4% of the total deliveries were premature. Among these reviewed cases, one maternal death and one neonatal death were also reported following COVID-19 infection. The birth weight of the babies was normal in most cases, although 17.9% of the newborns had low birth weight (LBW). |

(*Continued*)

**Table 1.** (Continued)

| Authors (2020) | Questions (Q) | Search Date[#] | (Q) N studies | (Q) N women | (Q) N neonates | AMSTAR-2 (1 to 16)[*] | Overall confidence | Main findings |
|---|---|---|---|---|---|---|---|---|
| Kotlyar [24] | 4 | 5/28/20 | 38 | NA | 936 | 7 | Critically low | A pooled proportion of 3.2% (95% confidence interval, 2.2–4.3) for vertical transmission. Severe acute respiratory syndrome coronavirus two viral RNA testing in neonatal cord blood was positive in 2.9% of samples (1/34), 7.7% of placenta samples (2/26), 0% of amniotic fluid (0/51), 0% of urine samples (0/17), and 9.7% of fecal or rectal swabs (3/31). Neonatal serology was positive in 3 of 82 samples (3.7%) (based on the presence of immunoglobulin M). |
| Li [32] | 2,3 | 2/6/20 | 13 | 19 | NA | 7 | Critically low | The clinical symptoms such as fever and cough in children with SARS infection are similar to that of adult patients |
| Martins [4] | 4 | 4/21/20 | 8 | 24 | 24 | 5 | Critically low | Most pregnant women had a cesarean delivery (91.7%) and two neonates had low birthweight (< 2 500 g). placental tissues and breast milk showed negative results for the presence SARS-CoV-2 by RT-PCR test. |
| Matar [25] | 2 | 4/30/20 | 24 | 136 | 3 (94), 4 (136) | 7 | Critically low | Most common symptoms were fever (62.9%) and cough (36.8%). Laboratory findings included elevated C-Reactive protein (57%) and lymphocytopenia (50%). Ground-glass opacity was the most common radiological finding (81.7%). Most patients were delivered via a cesarean delivery with a rate of 76.3% (95%CI 65.8–.84.2%). Thirty-one of 94 neonates were delivered preterm (<37 weeks). In all cases, the amniotic fluid, placenta, and umbilical cord samples all tested negative for SARS-CoV-2, while 2 neonates had RT-PCR–confirmed SARS-CoV-2 infection. |
| Melo [26] | 2 | 3/1/20 | 38 | 279 | NA | 9 | Critically low | The main reported laboratory findings were lymphopenia, elevated C-Reactive Protein (CRP), Amino alanine transferase (ALT), and Aspartate amino transferase (AST). In all symptomatic cases, chest Computerized Tomography (CT) scans were abnormal. Their signs and symptoms were all similar to the non-pregnant population. |
| Mirbeyk [47] | 4 | NA | 2 (17–37), 4 (37) | 364 | 219 | 8 | Critically low | 17 studies examined samples of the placenta, breast milk, umbilical cord, and amniotic fluid, and all tested negative except one amniotic fluid sample. Most mothers described mild to moderate manifestations of COVID-19. Of 364 pregnant women, 25 were asymptomatic at the time of admission. The most common symptoms were fever (62.4%) and cough (45.3%). Positive SARS-CoV2 test in neonates: 11/219 |
| Muhidin [48] | 3 | 3/19/20 | 9 | 89 | NA | 11 | Critically low | The main reported laboratory findings were lymphopenia, elevated C-Reactive Protein (CRP), Amino alanine transferase (ALT), and Aspartate aminotransferase (AST). In all symptomatic cases, chest Computerized Tomography (CT) scans were abnormal. |
| Mullins [33] | 4 | 7/12/1905 | 18 | 19 | 20 | 7 | Critically low | Delivering babies, 3 (16%) were asymptomatic, 1 (5%) was admitted to ICU and no maternal deaths have been reported. Deliveries were 17 by caesarean section, 2 by vaginal delivery, 8 (42%) delivered preterm. There was one neonatal death. |

(*Continued*)

**Table 1.** (Continued)

| Authors (2020) | Questions (Q) | Search Date# | (Q) N studies | (Q) N women | (Q) N neonates | AMSTAR-2 (1 to 16)* | Overall confidence | Main findings |
|---|---|---|---|---|---|---|---|---|
| Mustafa [27] | 4 | 4/2/20 | 6 | NA | 57 | 6 | Critically low | Vertical transmission and virus shedding in breast milk are yet to be established. |
| Panahi [72] | 4 | 3/30/20 | 14 | NA | 13 | 10 | Critically low | Vertical transmission of SARS-CoV-2 through placenta and its short-term and long-term harm to offspring is still unclear. Fetal distress, premature labor, respiratory distress and even death. One case with multiple organ damage and rapid disease changes like adults. One case asymptomatic despite high viral load |
| Pettirosso [49] | 2, 3 | 5/23/20 | 54 | 3830 | 655 | 11 | Critically low | Asymptomatic infection occurred in 43.5–92% of cases. Fever was the most common sign, occurring in 10–100% of cases both at admission and postpartum. 19 of a total 655 neonatal nasopharyngeal swabs were SARS-CoV-2 positive by rtPCR across ten studies. SARS-CoV-2 was not identified in any of the 45 breastmilk samples studies. |
| Rahman [50] | 4 | NR | 8 | NA | NA | 8 | Critically low | The first case of possible vertical transmission of COVID-19 was reported in March 2020. Other cases similarly showed neonates to be positive for the COVID-19 virus, shortly after birth, while the amniotic fluid, cord blood, breast milk after the first lactation of their mothers were negative of the virus. Not all neonates of COVID-19-positive mothers acquired the disease. The potential of vertical transmission of COVID19 should not be ruled out |
| Raschetti [77] | 4 | 8/30/20 | 74 | NA | 176 | 11 | Critically low | We report that 70% and 30% of infections are due to environmental and vertical transmission, respectively. Our analysis shows that 55% of infected neonates developed COVID-19; the most common symptoms were fever (44%), gastrointestinal (36%), respiratory (52%) and neurological manifestations (18%), and lung imaging was abnormal in 64% of cases. |
| Rodríguez-Blanco [64] | 3, 4 | 4/14/20 | 20 | 3 (79–102), 4 (79) | | 10 | Critically low | All three coronaviruses produce pneumonia with very similar symptoms, being milder in the case of SARSCoV2. Fever (75.5%) and pneumonia (73.5%) were the most frequent symptoms in infected pregnant women. The most frequent obstetric complications were the threat of premature delivery (23.5%) and caesarean section (74.5%). No vertical transmission detected. |
| Rostami [78] | 2 | 5/25/20 | 71 | 28 | NA | 7 | Critically low | D-dimer levels were found to be higher in non-COVID-19 pneumonia patients than COVID-19 patients. |
| Segars [65] | 3 | 4/6/20 | 79 | 162 | 162 | 8 | Critically low | Coronavirus Disease 2019 infection may affect adversely some pregnant women and their offspring. |
| Shi [28] | 2 | 4/20/20 | 11 | 173 | | 6 | Critically low | The incidence of elevated D-dimer was 82% (95% CI: 75–89%), elevated neutrophil count was 81% (69–91%), elevated C-reactive protein was 69% (58–79%), and decreased lymphocyte count was 59% (41–75%) |
| Singh [76] | 4 | 6/4/20 | 62 | NA | NA | 9 | Critically low | There is evidence of significant placental pathology in SARS-CoV-2 infection, but it is unclear what effects there may be for early pregnancy. |

(Continued)

**Table 1.** (Continued)

| Authors (2020) | Questions (Q) | Search Date# | (Q) N studies | (Q) N women | (Q) N neonates | AMSTAR-2 (1 to 16)* | Overall confidence | Main findings |
|---|---|---|---|---|---|---|---|---|
| Smith [35] | 3 | NA | 9 | 92 | 60 | 12 | Low | COVID-19-positive pregnant women present with fewer symptoms than the general population and may be RT-PCR negative despite having signs of viral pneumonia. The incidence of preterm births, low birth weight, C-section, NICU admission appear higher than the general population. |
| Soheili [29] | 2,3 | 4/1/20 | 11 | 177 | NA | 10 | Critically low | The most common signs and symptoms in pregnant women are fever, myalgia, increased CRP, increased LFT and Lymphopenia. All the pregnant women admitted to the hospital had radiological features of COVID-19 pneumonia in CT scan or CXR. The pooled prevalence of neonatal mortality, lower birth weight, stillbirth, premature birth, and intrauterine fetal distress in women with COVID 19 were 4% (95%Cl 1–9%), 21% (11–31%), 2% (1–6%), 28% (12–44%), and 15% (4–26%); respectively |
| Teles Abrao Trad [51] | 4 | 3/25/20 | 16 | 155 | 2 (118), 4 (95) | 8 | Critically low | Placenta, amniotic fluid, umbilical cord blood, breastmilk, gastric juice, urine, and feces were all screened for SARS-CoV-2 in different studies and were reported as negative, suggesting a possible lack of vertical transmission. Additionally, one patient who tested negative for SARS-CoV-2 PCR had positive SARS CoV-2 IgM and IgG. Hence, the possibility of vertical transmission is inconclusive at this point. |
| Thomas [52] | 4 | 5/7/20 | 18 | 157 | 2 (160), 4 (81) | 7 | Critically low | The next most commonly observed symptom was cough in 27 patients (40%). Symptom onset typically occurred before delivery in 44 patients (66%), after delivery in 21 patients (31%), and on the day of delivery in 2 patients (3%). Preterm birth, defined as gestational age less than 37weeks old, was observed in 24 (20%) neonates. Amongst 81 (69%) neonates who were tested for SARS-CoV2, 5 (6%) had a positive result. However, amongst these 5 neonates, the earliest test was performed at 16 h after birth, and only 1 neonate was positive when they were later re-tested |
| Trevisanuto [69] | 1 | 5/1/20 | 26 | NA | 44 | 12 | Critically low | Most neonates with SARS-CoV-2 infection were asymptomatic or presented mild symptoms, generally were left in spontaneous breathing and had a good prognosis after median 10 days of hospitalization. |
| Trippella [36] | 3, 2 | 4/18/20 | 37 | 275 | 4 (248), 4 (275) | 14 | Low | The majority of pregnant women presented with mild to moderate symptoms. SARS-CoV-2 infection in pregnant women appeared associated with mild or moderate disease in most cases, with a low morbidity and mortality rate. The outcomes of neonates born from infected women were mainly favorable. Out of the 191 tested neonates, 175 (92%) were negative. |

(*Continued*)

**Table 1.** (Continued)

| Authors (2020) | Questions (Q) | Search Date[#] | (Q) N studies | (Q) N women | (Q) N neonates | AMSTAR-2 (1 to 16)[*] | Overall confidence | Main findings |
|---|---|---|---|---|---|---|---|---|
| Trocado [53] | 3 | 3/20/20 | 8 | 95 | 51 | 11 | Critically low | The most common symptoms presented were fever (55%), cough (38%) and fatigue (11%). The most frequent pregnancy-related complications were premature rupture of membranes (PROM) (5%), fetal distress (14%), and postpartum fever (8%). Other related outcomes were gestational diabetes (3%), vaginal bleeding (3%), gestational hypertension (2%), placenta previa (2%), preeclampsia (1%), oligohydramnios (1%), polyhydramnios (1%) and low abdominal pain (1%). The mean birth weight of these 40 neonates was 2292g and 20% of the newborn infants had low birth weight (<2500g). No Apgar scores <5 at 1 min or <7 at 5 min were reported. |
| Turan [54] | 3 | 5/29/20 | 63 | 637 | 485 | 7 | Critically low | Most (76.5%) women experienced mild disease. |
| Uygun-Can [30] | 2 | 5/1/20 | 12 | 181 | NA | 8 | Critically low | Fever and cough are the most common symptoms in pregnant cases with SARS-CoV-2 infection, and 91.8% (95% CI: 76.7–99.9%) of RT-PCR results are positive. Abnormal CT incidence is 97.9% (95% CI: 94.2–99.9%) positive. No case was death. |
| Vakili [55] | 2 | NA | NA | NA | NA | 8 | Critically low | The most attracting and reliable markers in pregnant women were leukocytosis and elevated neutrophil ratio |
| Yang N [70] | 4 | 3/26/20 | 18 | 114 | 84 | 11 | Critically low | Fever (87.5%) and cough (53.8%) were the most commonly reported symptoms, followed by fatigue (22.5%), diarrhea (8.8%), dyspnea (11.3%), sore throat (7.5%), and myalgia (16.3%). There are reports of neonatal infection, but no direct evidence of intrauterine vertical transmission has been found. |
| Yang Z [71] | 4 | 4/20/20 | 22 | NA | 83 | 10 | Critically low | Three were confirmed with SARS-CoV-2 infection at 16, 36, and 72 hours after birth, respectively, by nasopharyngeal swab real-time polymerase chain reaction (RT-PCR) tests; another six had elevated virus-specific antibody levels in serum samples collected after birth, but negative RT-PCR test results. However, without positive RTPCR tests of amniotic fluid, placenta, or cord blood. |
| Yang Z [56] | 3 | 3/31/20 | 18 | 114 | NA | 9 | Critically low | Five case reports included sixteen breastfeeding mothers with COVID-19. The first case report showed that neonates of nine mothers with COVID-19 were isolated immediately after delivery and fed with formula. All samples of breast milk from 6 mothers and throat swab from their infants showed negative nucleic acid test results for SARS-CoV-2. The second case report described a familial cluster of SARS-CoV-2 infection. |
| Yee [31] | 2, 3 | 7/20/20 | 11 | 9032 | 338 | 7 | Critically low | Pregnant women with COVID-19 have relatively mild symptoms. However, abnormal proportions of laboratory parameters were similar or compared to general population. Fetal death and detection of SARS-CoV-2 were observed in about 2%, whereas neonatal death was found to be 0.4%. |

(*Continued*)

**Table 1.** (Continued)

| Authors (2020) | Questions (Q) | Search Date# | (Q) N studies | (Q) N women | (Q) N neonates | AMSTAR-2 (1 to 16)* | Overall confidence | Main findings |
|---|---|---|---|---|---|---|---|---|
| Yoon [57] | 4 | 4/15/20 | 28 | 223 | 221 | 11 | Critically low | Mothers with COVID-19 usually appeared with fever (42.3%), cough (31.8%); myalgia (21.4%); and dyspnea/short of breath (11.3%). 92.5% of confirmed women had pneumonia on CT scan. Leukocytosis was reported in 31.5% and lymphocytopenia was in 43.3%. C-reactive protein concentration (CRP) was elevated in 63.1%. Neonate (25%) had fetal distress and delivered prematurely with LBW (1580 g)22. All neonates showed pneumonia on chest imaging. Regarding pregnancy outcome, premature rupture of membrane was reported in 12.7% and preterm labor was reported in 22.7%. Fetal distress was reported in There were two still-births 15,39. Postpartum fever was reported in 34.3%. Reverse Transcription-PCR tests of the breast milk, placenta, amniotic fluids, and cord blood and maternal vaginal secretions were negative for SARS-CoV-2. Fetal death was reported in two cases. 48 of 185 newborns (25.9%) were born prematurely |
| Zaigham [58] | 3 | 4/1/20 | 18 | 108 | NA | 8 | Critically low | Fever (68%) and coughing (34%). Lymphocytopenia (59%) with elevated C-reactive protein (70%) |

**Question (Q): 1**. Prevalence, **2**. Signs & Symptoms, laboratory or image of child or mother, **3**. outcome (mortality, abortions, complications, etc.), **4**. Vertical transmission, **5** diagnostic accuracy, **6**. Effectiveness (cost & effect) of Interventions

**# Search date refers to the last search date if searches were performed at different times**

*****AMSTAR-2**: number of non-negative items out of 16 items, **AMSTAR-2 overall confidenc**e: Critically low, Low, Moderate, High

when compared to non-pregnant women of reproductive age with COVID-19, pregnant women with the disease were less likely to manifest fever (OR 0.43, 95%CI 0.22–0.85) and myalgia (OR 0.48, 95%CI 0.45–0.51).

## Maternal outcomes

Forty-four reviews [2, 4, 15–19, 21–23, 25, 26, 29, 31–37, 39–41, 43–49, 51–54, 56–65] reported at least one pregnancy outcome in pregnant women. The main findings are presented in **Table 3**, **Table 3 in S8 File**. The most frequently reported outcomes were C-section and preterm delivery.

The Allotey review [2] was chosen as the best review for reporting maternal outcomes according to predefined criteria (**Table 3**). It was reported that the prevalence of all-cause mortality was 0.63%, severe COVID-19 was 13%, admission to an intensive care unit (ICU) was 4%, and required invasive ventilation was 3%.

When compared with non-pregnant women of reproductive age with COVID-19, the reported odds of admission to the ICU was 1.62 (95%CI 1.33–1.96); and the reported odds of required invasive ventilation was 1.88 (95%CI 1.36–2.60).

Risk factors associated with severe COVID-19 were age (OR 1.78, 95%CI 1.25–2.55), high body mass index (OR 2.38, 95%CI 1.67–3.39), hypertension (OR 2.0, 95%CI 1.14–3.48), and pre-existing diabetes (OR 2.51, 95%CI 1.31–4.80).

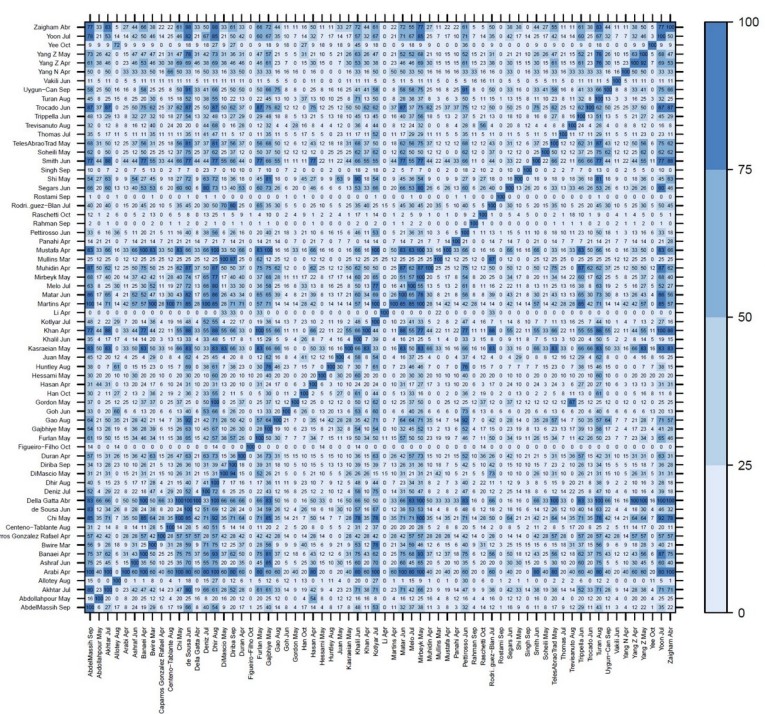

**Fig 2. Primary study overlap matrix (percentage) across included systematic reviews**[*]. [*]The figure displays the 66 included SRs in both axis and the percentage of its primary studies overlap between two SRs. Only primary articles with doi were included for this figure.

## Neonatal clinical presentation

Twenty-eight reviews [2, 15, 31, 33, 35–37, 39–42, 45, 47, 48, 51–54, 57, 60, 61, 63, 66–71] reported information on the clinical presentation in neonates born from pregnant women diagnosed with SARS-CoV-2 (**Table 2**, **Table 4 in S8 File**).

Respiratory distress syndrome, shortness of breath, and fetal distress were the most frequently reported moderate to severe presentations.

Laboratory data (confirmed cases through Polymerase chain reaction [PCR], elevated SARS-CoV-2 IgM, and IgG antibodies) and imaging data (radiographic pneumonia) were also reported.

The Figueiro-Filho review [39] was chosen as the best review for reporting neonatal clinical presentation according to predefined criteria (**Table 3**). This review involved almost 11,000 cases of COVID-19 and pregnancy in 15 different countries. The most frequent newborn complications were admission to neonatal ICUs (NICUs) (18.5%), prematurity complications (5.43%), respiratory distress syndrome (4.9%), and congenital abnormalities (3.3%). The rate of neonatal hospitalization < 2 days was 62.4% and > 7 days 11.8%.

## Neonatal outcomes

Forty-one studies [2, 16–19, 21–23, 25, 29, 31, 33–37, 39–41, 43–45, 47, 48, 51–54, 57, 58, 60–67, 69, 70, 72] presented neonatal outcomes of pregnant women diagnosed with SARS-CoV-2 (**Table 2**, **Table 5 in S8 File**). The most frequently reported neonatal outcomes were low birth weight and preterm delivery.

The Allotey review [2] was chosen as the best review for reporting neonatal outcomes according to predefined criteria (**Table 3**). The reported neonatal mortality prevalence was

**Table 2. Ranges of outcomes reported in the included systematic reviews\*.**

| Dimension | Outcome | # of studies | References | Range of mothers / neonates analyzed | Range of outcome |
|---|---|---|---|---|---|
| **Clinical presentation pregnant women** | Asymptomatic | 15 | [2, 15, 22, 23, 33, 35, 36, 39, 42, 43, 49, 51, 52, 54, 55] | 19 to 6598 | 4.8% to 41% |
| | Pneumonia | 10 | [2, 18, 22, 35–37, 41, 45, 47, 64] | 70 to 2577 | 6% to 100% |
| | Mild respiratory symptoms | 34 | [2, 4, 15, 16, 18, 19, 21–23, 29–32, 34–37, 39–47, 51–54, 56–58, 64] | 19 to 8560 | 20% to 78.94% |
| | Fever | 35 | [2, 15, 16, 18, 19, 21–23, 25, 29–32, 34–37, 39–47, 49, 51–53, 56–58, 64] | 19 to 8571 | 27.6% to 100% |
| | Headache | 8 | [18, 21, 23, 34, 39, 41, 49, 54] | 161 to 3474 | 3.3% to 40.69% |
| | Dyspnea/shortness of breath | 30 | [2, 15, 16, 18, 21, 23, 25, 29, 30, 34–37, 39–44, 46, 49, 51–54, 56–58, 64] | 32 to 1941 | 3.3% to 75.2% |
| | Fatigue/malaise | 20 | [16, 18, 23, 29, 31, 34–37, 39–43, 46, 49, 52, 53, 56, 58] | 35 to 680 | 6.45% to 30.49% |
| | Myalgia | 25 | [2, 15, 16, 18, 21, 23, 29, 31, 34, 36, 37, 39–46, 49, 52–54, 56–58, 64] | 19 to 8372 | 1% to 43.5% |
| | Diarrhea | 22 | [2, 15, 16, 21, 23, 25, 29, 31, 36, 37, 39, 41–43, 46, 49, 52–54, 56, 58, 64] | 35 to 8310 | 0% to 15.6% |
| | Mechanical ventilation/ IUC | 17 | [2, 15, 21, 23, 32–35, 37, 39–41, 44, 45, 51, 54, 64] | 19 to 10713 | 1.4% to 83.6% |
| | Death | 21 | [2, 16, 18, 21–23, 25, 32, 34–37, 41, 42, 44, 45, 47, 52, 54, 58, 64] | 19 to 11580 | 0% to 15.78% |
| | Raised reactive C protein | 17 | [2, 18, 23, 25, 28, 29, 31, 36, 37, 39, 41, 42, 48, 55, 57, 58] | 30 to 592 | 27.59% to 96% |
| | Lymphocytopenia | 24 | [2, 18, 19, 21–23, 25, 28, 29, 31, 32, 34–37, 39, 41, 42, 48, 51, 54, 55, 57, 58] | 28 to 780 | 33.6% to 80% |
| | Leukocytosis/ Neutrophilia | 9 | [2, 18, 28, 41, 42, 51, 54, 55, 57] | 24 to 251 | 8.8% to 81% |
| | Elevated ALT or AST | 13 | [2, 18, 21, 23, 25, 36, 37, 39, 41, 42, 44, 54, 55] | 25 to 491 | 8.2% to 38.6% |
| | Signs of pneumonia X Rays or CT | 10 | [2, 19, 21, 32, 37, 41, 44, 51, 54, 57] | 19 to 1968 | 7.1% to 99% |
| **Maternal outcomes** | Preterm delivery | 33 | [2, 16–19, 21–23, 25, 29, 31, 33–36, 39, 40, 43, 45, 47, 51–54, 57, 60–63, 65–67, 70] | 19 to 13118 | 14.3 to 63.8% |
| | Vaginal delivery | 22 | [18, 21, 22, 29, 33, 35–37, 39–41, 43, 48, 51–54, 58, 60, 62, 64, 70] | 19 to 1119 | 1% to 38.6% |
| | C-section | 35 | [2, 4, 16–19, 21–23, 25, 26, 29, 32–37, 39–41, 43, 46–48, 51–54, 58, 60, 62–64, 70] | 12 to 1125 | 23% to 95.8% |
| | Preeclampsia | 14 | [16–18, 21, 29, 36, 37, 41, 43, 45, 48, 51–53] | 10 to 381 | 0% to 26% |
| | Stillbirth | 21 | [2, 17, 21–23, 29, 34, 36, 37, 41, 43–46, 48, 49, 54, 57, 58, 60, 70] | 13 to 663 | 0% to 8% |
| | Gestational diabetes | 7 | [36, 37, 41, 45, 51–53] | 44 to 369 | 0.7% to 29% |
| | Hypertension | 8 | [36, 37, 41, 43, 51–53, 64] | 44 to 275 | 2% to 11.4% |
| | Premature rupture of membranes | 18 | [17, 18, 21, 22, 31, 33, 36, 40, 41, 43, 45, 49, 51–53, 57, 62, 64] | 31 to 714 | 1% to 41% |
| | Miscarriage or abortion | 8 | [18, 21, 32, 37, 44, 49, 54, 65] | 10 to 743 | 0.5% to 14.5% |
| | Intrauterine growth retardation | 5 | [18, 32, 61, 64, 65] | 13 to 162 | 0% to 23% |
| | Fetal distress | 22 | [2, 15–19, 22, 23, 25, 29, 31, 35, 40, 41, 45, 48, 51–54, 65, 70] | 18 to 369 | 0.1% to 46.7% |

(*Continued*)

**Table 2.** (Continued)

| Dimension | Outcome | # of studies | References | Range of mothers / neonates analyzed | Range of outcome |
|---|---|---|---|---|---|
| **Neonatal clinical presentation** | Asymptomatic | 6 | [31, 33, 35, 41, 63, 69] | 20 to 86 | 16% to 93.2% |
| | Fever | 11 | [15, 31, 36, 37, 40, 48, 60, 63, 66, 67, 69] | 10 to 222 | 0% to 50% |
| | Gastrointestinal symptoms | 9 | [31, 36, 40, 48, 52, 60, 63, 68, 69] | 10 to 160 | 1% to 8.1% |
| | Shortness of breath | 10 | [31, 37, 40, 48, 52, 60, 63, 66, 67, 69] | 10 to 222 | 0% to 60% |
| | Respiratory distress syndrome | 9 | [31, 36, 39, 41, 45, 51, 54, 60, 67] | 86 to 576 | 0.8% to 11.1% |
| | Mild respiratory symptoms | 5 | [36, 37, 60, 67, 69] | 34 to 222 | 0.8% to 20% |
| | Elevated SARSCov2 IgM | 6 | [31, 42, 45, 61, 69, 70] | 8 to 493 | 0.6% to 17.6% |
| | Elevated SARSCov2 IgG | 5 | [31, 42, 45, 61, 70] | 8 to 493 | 0.6% to 35% |
| | Radiology pneumonia | 7 | [31, 36, 37, 45, 60, 67, 69] | 21 to 369 | 0% to 71% |
| **Neonatal outcomes** | Mortality | 40 | [2, 16–19, 21–23, 25, 29, 31, 33–37, 39–41, 43–45, 47, 48, 51–54, 57, 58, 60, 61, 63–67, 69, 70, 72] | 10 to 1728 | 0% to 9.2% |
| | Low birth weight (rate) | 11 | [21, 23, 29, 35, 39, 48, 51, 53, 57, 65, 70] | 21 to 598 | 5.3% to 42.9% |
| | Small for Gestational age | 9 | [31, 41, 48, 52, 54, 57, 60, 61, 66] | 10 to 479 | 1.25% to 20% |
| | Preterm (<37 weeks) | 33 | [2, 16–19, 21–23, 25, 29, 31, 33–36, 39, 40, 43, 45, 47, 51–54, 57, 60–63, 65–67, 70] | 10 to 1872 | 2% to 68.8% |
| | Low Apgar (<7) | 11 | [17, 18, 21, 34–36, 54, 60, 64, 65, 67] | 17 to 361 | 0% to 18.76% |
| | Admission to NICU | 18 | [2, 17, 18, 21, 23, 25, 33–35, 37, 39, 43, 45, 47, 51, 54, 60, 69] | 10 to 1348 | 1.6% to 76.9% |
| **Vertical transmission** | Placenta | 15 | [21, 23–27, 37, 41, 42, 47, 49, 54, 57, 62, 76] | 1 to 63 | 0 to 12.7% |
| | Amniotic | 15 | [21, 23–27, 35, 37, 41, 47, 52, 54, 57, 62, 63] | 3 to 81 | 0 to 11.1% |
| | Cord blood | 12 | [21, 23, 25–27, 35, 37, 47, 49, 52, 54, 57] | 4 to 81 | 0 to 14.3% |
| | Breastfeeding or breast milk | 21 | [4, 21, 23, 24, 26, 27, 36–38, 47, 49, 51, 52, 54, 56, 57, 59, 62, 64, 70, 71] | 6 to 82 | 0% to 19.8% |
| | Respiratory droplets | 21 | [4, 15, 21, 24, 26, 27, 35–38, 41, 45, 49, 51, 54, 56, 57, 63, 68, 70, 71] | 4 to 889 | 0% to 70.7% |
| | SARS-CoV-2 cases in neonates | 44 | [4, 15, 17, 18, 21, 23–27, 31, 33, 35–39, 41, 42, 44, 45, 47, 49, 51, 52, 54, 56–64, 66–68, 70, 71, 73–75] | 4 to 1116 | 0% to 27.3% |

* The S8 File presents the systematic review level data by research question, that were aggregated in Table 2.

**CT**: computed tomography

0.34%, and neonatal admission to NICUs was 25% (95%CI 14%-37%). The neonates born to mothers diagnosed with COVID-19 presented an odds ratio of admission to critical care unit equal to 3.13 (95%CI OR 2.05–4.78) compared with those born to mothers without the disease. Finally, the rate of overall preterm birth reported among pregnant women diagnosed with COVID-19 was 17% (95%CI 13%-21%). No significant findings were observed for other neonatal outcomes.

## Vertical transmission

Forty-six reviews [4, 15, 17, 18, 20, 21, 23–27, 31, 33, 35–39, 41, 42, 44, 45, 47, 49–52, 54, 56–64, 66–68, 70, 71, 73–76] reported mother-to-child SARS-CoV-2 transmission (**Table 2, Table 6 in S8 File**). Most studies only reported the proportion of infants' positive cases without evaluating breast milk or congenital/perinatal transmission. In the reviews reporting breast milk or congenital/perinatal transmission the sample analyzed was generally small for these outcomes.

The Centeno-Tablante review [38] was chosen as the most appropriate review to answer this question according to predefined criteria (**Table 3** and **S10 File**). This review included 37 papers with a total of 889 infants. Of the 72 infected mothers whose breast milk samples were laboratory-confirmed to contain the COVID-19 antigen, 14 infants were found to be infected with COVID-19. Eight of the twenty-three infants that were breastfed were infected, two of the eighteen infants that received a breast milk substitute were infected, two of four infants that received mixed feeding were infected, and two of the twenty-three infants that did not report

**Table 3. Key messages of prioritized systematic reviews.**

| Question Author year | Key messages | № of participants (studies)# | Certainty of the evidence (GRADE) |
|---|---|---|---|
| **What is the clinical presentation of pregnant women with COVID-19?**[S] Allotey 2020 [2] | Compared with non-pregnant women of reproductive age with COVID-19, pregnant women may be less likely to manifest symptoms. Fever, cough, dyspnea and ageusia may be the most frequent symptoms. | 310 to 8328 (3 to 29 studies) | ⊕⊕◯◯ Low |
| | Raised C-reactive protein level, lymphopenia, raised white cell count and raised procalcitonin level may be the most frequent laboratory findings. | 251 to 780 (5 to 15 studies) | ⊕⊕◯◯ Low |
| | Ground glass appearance may be the most frequent radiological finding | 387 to 1960 (10 to 20 studies) | ⊕⊕◯◯ Low |
| **What are the outcomes of pregnant women with COVID-19?** Allotey 2020 [2] | Compared with pregnant women without COVID-19, pregnant women with COVID-19 may have more admissions to intensive care units | 1121 (1 study) | ⊕⊕◯◯ Low |
| | Compared with pregnant women without COVID-19, pregnant women with COVID-19 may have more deaths of any cause, preterm births, and cesarean sections. However, the evidence is very uncertain. | 339 to 2167 (1 to 3 studies) | ⊕◯◯◯ Very low |
| **What is the clinical presentation of infants born from pregnant women with COVID-19?** Figueiro-Filho [39] | The most frequent newborn complications were respiratory distress syndrome (4.86%), sepsis (0.4%), congenital abnormalities (3.3%), prematurity (5.43%) and admission to neonatal intensive care units-NICU (18.45%) | 241 to 992 (1 to 5 studies) | ⊕◯◯◯ Very low |
| | The rate of neonatal hospitalization < 2 days was 62.4%, 3–7 days 26.5% and > 7 days 11.8%. | 245 (1 study) | ⊕◯◯◯ Very low |
| **What are the neonatal outcomes born from pregnant women with COVID-19?** Allotey 2020 [2] | Compared with neonates born from women without COVID-19, the neonates born from women with COVID-19 may have more admissions to intensive care units | 1121 (1 study) | ⊕◯◯◯ Low |
| | Compared with neonates born from women without COVID-19, the neonates born from women with COVID-19 may have more deaths, and fetal distress and no important differences in abnormal Apgar score at 5 minutes. However, the evidence is very uncertain. | 376 to 1121 (1 study) | ⊕⊕◯◯ Very low |
| **Do mothers transmit SARS-CoV-2 infection to their offspring through breastfeeding?**[S] Centeno-Tablante 2020 [38] | Transmission via breastfeeding through other related bodily fluids, (i.e., droplet transmission or airborne transmission due to close contact with the infant or young child) could pose a risk to the infant. However, the evidence is very uncertain. | 77 children (37 studies) | ⊕◯◯◯ Very low |
| | SARS-CoV-2 transmission via breast milk is very uncertain and the risk of transmission via this route is estimated to be, at most, low. | 82 breast milk samples (37 studies) | ⊕◯◯◯ Very low |
| **Is there congenital transmission of SARS-CoV-2 between mothers and children?** Juan 2020 [37] | Apparently, vertical mother-to-infant transmission is low. However, there is not enough good quality data to draw unbiased conclusions. | 155 children (19 studies) | ⊕◯◯◯ Very low |
| | Congenital (umbilical cord blood and amniotic fluid) transmission of SARS-CoV-2 is highly uncertain and the risk of transmission by this route is estimated to be low at best. However, the evidence is very uncertain. | 32 cord blood and 34 amniotic samples (7 studies) | ⊕◯◯◯ Very low |

[S] The policy briefs for these systematic reviews are available in S9 and S10 Files.

# The number studies and participants vary across different specific outcomes.

on feeding practice were infected. Regarding congenital/perinatal transmission cases, the Juan review [37] was deemed to be the most appropriate review (**Table 3**). This review included 24 studies, case series, and case reports, including a total of 155 neonates. Ninety neonates were tested for COVID-19, of which three were positive. The review also evaluated the presence of SARS-CoV-2 in amniotic fluid (1/32), umbilical cord blood (0/34), and placenta (1/3). While there was no vertical mother-to-child transmission, additional good-quality studies are needed to determine whether vertical transmission is possible.

The key messages from the prioritized systematic reviews (by the most current search date, the more significant number of included studies, and greater adequacy to address the outcomes) are presented in **Table 3**.

## Discussion

This systematic review of SRs integrated the most consolidated evidence synthesis regarding the effects of COVID-19 on maternal and neonatal health.

Most SRs (92.4%) were classified as "critically low" in overall confidence, using the AMSTAR-2 tool, likely due to the urgent demand of information for this hot topic. For the prioritized systematic reviews, we also used the ROBIS tool (one was classified as "low risk of bias" [2], one as "unclear risk of bias" [38] and the other two as "high risk of bias" [37, 39]) and the GRADE approach for each of their outcomes. The certainty of evidence was rated as "low" to "very low" due to study design, risk of bias, inconsistency and/or imprecision.

The COVID-19 related symptoms manifest from one third to two thirds less often in pregnant women than in non-pregnant women of reproductive age [2]. While testing for SARS-CoV-2 in non-pregnant women is usually based on symptoms or contact history, testing in pregnant women is generally done for reasons that might not be related to COVID-19. Affected women were at higher risk of requiring admission to an ICU or invasive ventilation. Pregnant women with COVID-19 are also at an increased risk of receiving cesarean sections, delivering preterm and their babies being admitted to a NICU. Higher age, higher body mass index, and pre-existing comorbidities might be associated with severe disease [2]. Stillbirth and neonatal death rates are low in women with suspected or confirmed COVID-19. However, this evidence is based on a few large comparative studies. The substantial heterogeneity identified could be related to using different sampling techniques, the different sampling techniques, and the differential baseline risk of participants [2].

Most infected neonates were asymptomatic. The most frequent symptoms were fever (0–50%) or mild respiratory symptoms [39]. Low birth weight and preterm birth were the most frequently reported neonatal outcomes. The neonates born to mothers diagnosed with COVID-19 presented an odds three times higher admission to a NICU compared with those born to mothers without the disease [2]. The risk of congenital transmission [37] or transmission via breast milk [38] is estimated to be low to very low, but there is a higher risk of transmission due to close contact by droplet or airborne transmission.

The high rate of asymptomatic presentation in pregnant women with COVID-19 [2] may be explained by the screening strategy for COVID-19 and the low thresholds for testing during pregnancy. Even detecting more pregnant women with mild disease, higher admissions to the ICU, or invasive ventilation were observed when they are compared with non-pregnant women of reproductive age with COVID-19 [79]. A Swedish study also suggested an more admissions to an ICU and higher requirement of invasive ventilation in pregnant women than non-pregnant women [80].

Similar to the general population, pre-existing comorbidities seemed to be risk factors for severity of COVID-19 in pregnancy [81]. Adverse outcomes related to COVID-19 were not

found to be higher in women at the third trimester nor in multiparous ones—but the existing sample sizes are not large (less than 300 women). Chronic hypertension and pre-existing diabetes were associated with maternal death in pregnant women with COVID-19, and both are recognized risk factors in the general population. The low numbers of studies does not allow to stablish the cause of death for these women. A slight increase in rates of preterm birth in pregnant women with COVID-19 was observed when compared to those without the disease. These preterm births could be medically indicative, since rates of spontaneous preterm births in affected women were similar to those before the pandemic. More than 60% of pregnant women underwent cesarean section in the non-comparative studies. This is three times the global rate of cesarean sections worldwide [82], and deserves future research.

Surely, the precision will improve as more data is published. The overall rates of stillbirths and neonatal mortality are not observably higher than the background rates. The indicators for admissions to the NICU, of about 25% of neonates from affected mothers were not reported. Countries' regulations on isolation of exposed infants to the virus may have influenced these rates.

Sixty-seven percent of newborns delivered by mothers with COVID-19 antibodies had SARS-CoV-2 IgG, but not IgM antibodies [83]. This finding against vertical transmission is consistent with our own findings of low to very low risk of this mechanism of congenital transmission.

To our knowledge, there is only one overview of SRs published that reports maternal and perinatal outcomes related to COVID-19 and pregnancy [84], including 52 SRs. This overview searched studies until September 2020 and did not include 14 SRs that were found in our overview, probably explaining the lower level of overlap observed in our study. The authors did not assess the quality of reporting of each SR, but they assessed the risk of bias of each included SR using the ROBIS tool [11]. The high risk of bias identified in this overview is consistent with the "critically low" confidence presented in our study by applying the AMSTAR-2 tool [10].

Initial studies involved women from China, but later in 2020, reports came also from regional or national data from European countries and Latin America. The study design has also changed from small case series and case reports to extensive observational data, with recent studies also being comparative. Variations in the criteria for doing testing (symptoms-based, close contact) and sampling methodologies explain differences in the prevalence of COVID-19. Moreover, the findings only apply to women attending the hospital for any reason. The true prevalence of COVID-19 in pregnancy is likely to be lower when all pregnant women are included.

## Strengths and limitations

This overview has several strengths. First, we followed sound methodology to conduct the present overview of systematic reviews. Second, we included systematic reviews without language restrictions. Third, we adhered to rigorous quality appraisal for the conduction of systematic reviews (AMSTAR-2 tool), which was independently assessed by pairs of reviewers and discrepancies solved by consensus. We summarized and critically appraised an important amount of evidence that is relevant to health decision-making (See examples of policy briefs in S9 and S10 Files) and highlighted evidence gaps that could guide future research. Finally, we conducted a sensitive and comprehensive search strategy to reduce the risk of missing relevant studies. We synthesized the results of all relevant SRs, highlighting their overlap through a matrix of primary studies by SR, and we also selected the best SR that answered a specific question according to pre-defined criteria of relevance, comprehensiveness, data update, and quality. We presented these SRs through a summary of the findings tables with the certainty of

evidence according to the GRADE approach. Our review integrated the evidence generated by different independent groups, which could improve the robustness of our findings.

Our study is not exempt from limitations. A main limitation is that the last search was run in October 2020. This is due to the time needed to perform a thorough SR. The general confidence of the included SRs was "critically low", and the certainty of evidence was "low" to "very low". Additionally, there is a scarcity of data comparing pregnant women with non-pregnant women or comparing pregnant women with and without COVID-19 [2]. We did not evaluate the risk of bias of the primary studies, nor did we undertake a pooled analysis by the outcome, as was originally stated in our protocol. Nevertheless, there was great heterogeneity of methods, study designs, and estimations that would preclude a meta-analysis.

## Implications for clinical practice and research

Although pregnant women with COVID-19 could be less symptomatic than the general population, the overall pattern is similar. However, the admission rates to ICUs and invasive ventilation in pregnant women with COVID-19 could be higher than non-pregnant women. Mothers with pre-existing comorbidities will need to be considered as a high-risk group for COVID-19, along with those who are obese and of advanced maternal age.

Physicians need to weight the need for antenatal visits against unnecessary exposure and use of virtual meetings whenever possible. Infected pregnant women with before-term gestations might need care in special facilities for these cases since the neonates born to mothers diagnosed with COVID-19 are at three times the risk of admission to NICUs compared with those born to mothers without the disease.

Further data are needed to assess robustly if pregnancy-related maternal and neonatal complications are increased in women with COVID-19 than those without the disease. Similarly, the association between other risk factors, such as ethnicity and pregnancy-specific risk factors such as preeclampsia and gestational diabetes on both COVID-19 related and pregnancy-related outcomes needs evaluation. Preeclampsia was reported to be associated with severe COVID-19 in small studies, but requires a further assessment as the clinical presentation of severe preeclampsia could mimic worsening COVID-19 [85]. Robust registers of pregnancy data by trimester of exposure are essential to determine the effects of COVID-19 on early maternal and neonatal outcomes.

Systematic reviews of RCTs are the highest quality evidence to inform guidelines, and poor-quality systematic reviews will still directly impact on clinical care. Despite urgency for evidence, systematic reviews still need to adhere to the highest standards of reporting and conduction, more so in the presence of pre-prints, reports, and media statements. Primary studies will need to explicitly state if duplicate data have been included to avoid double counting participants in evidence synthesis. Individual participant data meta-analysis and network meta-analysis of the emerging cohorts are critical to evaluate clinical manifestations and outcomes by underlying risk factors and also to determine the differential effects of interventions to reduce the complication rates.

## Supporting information

**S1 File. Preferred reporting items for overviews of systematic reviews.**
(DOCX)

**S2 File. Outcomes of interest.**
(DOCX)

**S3 File. Search strategy.**
(DOCX)

**S4 File. Excluded studies & exclusion reasons.**
(DOCX)

**S5 File. Primary study list by systematic review.**
(XLSX)

**S6 File. Primary study overlap matrix, in absolute numbers, across included systematic reviews.**
(XLSX)

**S7 File. Quality assessment of systematic reviews.**
(DOCX)

**S8 File. Extracted data at systematic review level by the research question.**
(DOCX)

**S9 File. Policy brief of clinical presentation of pregnant women with COVID-19.**
(DOCX)

**S10 File. Policy brief of transmission of SARS-CoV-2 through breastfeeding.**
(DOCX)

## Acknowledgments

We thank Erin Goucher for her help with the English edition of the manuscript.

## Author Contributions

**Conceptualization:** Agustín Ciapponi, Ariel Bardach, Mabel Berrueta, Xu Xiong, Agustina Mazzoni, Pierre Buekens.

**Data curation:** Agustín Ciapponi, Ariel Bardach, Daniel Comandé, Mabel Berrueta, Fernando J. Argento, Federico Rodriguez Cairoli, Natalia Zamora, Victoria Santa María, Pierre Buekens.

**Formal analysis:** Agustín Ciapponi, Ariel Bardach, Mabel Berrueta, Fernando J. Argento, Federico Rodriguez Cairoli, Natalia Zamora, Victoria Santa María, Sabra Zaraa, Agustina Mazzoni, Pierre Buekens.

**Funding acquisition:** Pierre Buekens.

**Investigation:** Agustín Ciapponi, Daniel Comandé, Mabel Berrueta, Fernando J. Argento, Federico Rodriguez Cairoli, Natalia Zamora, Victoria Santa María, Xu Xiong, Sabra Zaraa, Agustina Mazzoni.

**Methodology:** Agustín Ciapponi, Ariel Bardach, Daniel Comandé, Mabel Berrueta, Xu Xiong, Sabra Zaraa, Agustina Mazzoni.

**Project administration:** Agustín Ciapponi, Mabel Berrueta, Pierre Buekens.

**Resources:** Pierre Buekens.

**Supervision:** Agustín Ciapponi, Ariel Bardach, Mabel Berrueta, Pierre Buekens.

**Validation:** Agustín Ciapponi, Ariel Bardach.

**Writing – original draft:** Agustín Ciapponi, Ariel Bardach, Mabel Berrueta, Fernando J. Argento, Federico Rodriguez Cairoli, Natalia Zamora, Victoria Santa María, Xu Xiong, Sabra Zaraa, Agustina Mazzoni, Pierre Buekens.

**Writing – review & editing:** Agustín Ciapponi, Ariel Bardach, Daniel Comandé, Mabel Berrueta, Fernando J. Argento, Federico Rodriguez Cairoli, Natalia Zamora, Victoria Santa María, Xu Xiong, Sabra Zaraa, Agustina Mazzoni, Pierre Buekens.

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
