## [Decision Letter · Decision Letter 0]

27 May 2021

PONE-D-21-14273

COVID-19 and pregnancy: An umbrella review of clinical presentation, vertical transmission, and maternal and perinatal outcomes

PLOS ONE

Dear Dr.Ciapponi,

Thank you for submitting your manuscript to PLOS ONE. After careful consideration, we feel that it has merit but does not fully meet PLOS ONE’s publication criteria as it currently stands. Therefore, we invite you to submit a revised version of the manuscript that addresses the points raised during the review process.

We look forward to receiving your revised manuscript.

Kind regards,

Linglin Xie

Academic Editor

PLOS ONE

Journal Requirements:

2. Please update search and analysis to include relevant publications since October 2020.

3. Thank you for submitting the above manuscript to PLOS ONE. During our internal evaluation of the manuscript, we found significant text overlap between your submission and the following previously published work:

https://www.bmj.com/content/370/bmj.m3320

The text that needs to be addressed involves several paragraphs throughout the Discussion section. Please revise the manuscript to rephrase the duplicated text, cite your sources, and provide details as to how the current manuscript advances on previous work. Please note that further consideration is dependent on the submission of a manuscript that addresses these concerns about the overlap in text with published work.

Reviewers' comments:

Reviewer's Responses to Questions

**Comments to the Author**

1. Is the manuscript technically sound, and do the data support the conclusions?

Reviewer #1: Yes

2. Has the statistical analysis been performed appropriately and rigorously? 

Reviewer #1: N/A

3. Have the authors made all data underlying the findings in their manuscript fully available?

Reviewer #1: Yes

4. Is the manuscript presented in an intelligible fashion and written in standard English?

Reviewer #1: Yes

5. Review Comments to the Author

Reviewer #1: • Overall, the authors provided a great deal of detail and transparency. The supplements provided additional information that increased this transparency and, in my opinion, adequately supported that data reported.

• Table 1: I think an expanded caption will help the reader with reviewing this table that may include the breakdown again of the AMSTAR-2 rating again. Additionally, I feel the “Search Data” column title may be clarified in the caption that this is the end search date (I believe that is what I determined that to be).

• Table 2: Again, an enhanced caption that references S8 (in addition to what you included in the text). When reviewing this table, I wanted to know which SRs reported which data. While this is highlighted in the body of the paper, including it in the caption for the table would be helpful.

• Table 2: Column title Range of Outcome has an asterisk next to it. Were you going to include additional information related to this in a caption? I am assuming this is the range of incidence reported in the studies referenced, but this is not clear from the table and text.

• Lines 182-183 appear out of place. Was this to be included with one of the tables?

• Figure 2: In my version of the manuscript, this figure was blurred and not visible.

• The information in lines 185-186: This information is described in Table 1. Is that correct? If so, would reference the reader to Table 1. Also, should the references that are critically low, low and moderate be cited next to those AMSTAR-2 grades?

• Information in lines 323-327: Is this information from this review, reference #2 or other reference? May wish to clarify the source.

• Lines 367-368: Just to clarify, the review #84 includes 52 SRs? If so, may wish to clarify this statement.

• Line 399: through or thorough?

• I scanned the references included in this umbrella review. I came across the publication by Vakili et al. Does this article fully meet your inclusion criteria? To me, it does not appear to be a systematic review, but you can correct me if I'm wrong.

• Overall writing: I found that the writing was appropriate for publication, minus a few minor punctuation and writing errors (omitted commas, covid vs. COVID, proper nouns not consistently capitalized). However, I did notice some statements or sections that need clarification:

-Table 3, question 5: Clarify statement. I believe there is a word or two missing

-Lines 166-167: Clarify statement – Table 1 is included and excluded studies? Or is Table 1 included and S4 Table excluded studies?

Thanks! Good job and thanks for this very interesting publication.

6. PLOS authors have the option to publish the peer review history of their article (what does this mean?). If published, this will include your full peer review and any attached files.

Reviewer #1: No

---

## [Author Response · Author response to Decision Letter 0]

10 Jun 2021

Review Comments to the Author 

Reviewer #1: 

• Overall, the authors provided a great deal of detail and transparency. The supplements provided additional information that increased this transparency and, in my opinion, adequately supported that data reported.

• Table 1: I think an expanded caption will help the reader with reviewing this table that may include the breakdown again of the AMSTAR-2 rating again. Additionally, I feel the “Search Data” column title may be clarified in the caption that this is the end search date (I believe that is what I determined that to be).

We agree with your suggestions. The addition to the Table 1 caption is highlighted below:

“Table 1. Main characteristics, research questions and findings of included systematic reviews”

In order to clarify the column title “search date” we added a footnote #: “# Search date refers to the last search date if searches were performed at different times”

• Table 2: Again, an enhanced caption that references S8 (in addition to what you included in the text). When reviewing this table, I wanted to know which SRs reported which data. While this is highlighted in the body of the paper, including it in the caption for the table would be helpful.

In order to address your proper suggestions, we added a footnote to the title *. “* The S8 presents the systematic review level data by research question, that were aggregated in Table 2.” We also added the references here.

• Table 2: Column title Range of Outcome has an asterisk next to it. Were you going to include additional information related to this in a caption? I am assuming this is the range of incidence reported in the studies referenced, but this is not clear from the table and text.

We had considered that it’s unnecessary additional information but we had omitted to delete the asterisk. Deleted now, thank you.

• Lines 182-183 appear out of place. Was this to be included with one of the tables?

This is just a relevant footnote for Fig 2 that must be presented separately:

• Figure 2: In my version of the manuscript, this figure was blurred and not visible.

Unfortunately, in the automatic pdf built is blurred but the image file meets the journal requirements. I am sure that the journal could process it to improve visibility. Although we present the overlapped number of SRs in each cell the most important for this heat map is to appreciate the general level of overlap suggested by the color pattern.

• The information in lines 185-186: This information is described in Table 1. Is that correct? If so, would reference the reader to Table 1. Also, should the references that are critically low, low and moderate be cited next to those AMSTAR-2 grades?

You are right. The overall quality, based on AMSTAR-2 it’s presented in Table 1 while S7 presents the quality assessment of systematic reviews also by each domain.

We included the following highlighted additions:

“Concerning the overall quality, based on AMSTAR-2 (see Table 1), most SRs were classified as "critically low" (n=61), four as "low"[2, 34-36], and only one as "moderate"[37].”

We considered that the 61 "critically low" could be found directly in Table 1. It would be too long to include all these reference.

• Information in lines 323-327: Is this information from this review, reference #2 or other reference? May wish to clarify the source.

You are right, it’s refers to reference 2. We also added the same reference at the end of the line 327 to clarify the source.

• Lines 367-368: Just to clarify, the review #84 includes 52 SRs? If so, may wish to clarify this statement.

It is correct. The overview (#84 Vergara-Merino 2021) includes 52 SRs. Overview refers to a systematic review of SRs. In order to clarify this concept we included the following highlighted addition: 

“To our knowledge, there is only one overview of SRs published that reports maternal and perinatal outcomes related to COVID-19 and pregnancy [84], including 52 SRs.”

• Line 399: through or thorough?

Changed to thorough. Thank you.

• I scanned the references included in this umbrella review. I came across the publication by Vakili et al. Does this article fully meet your inclusion criteria? To me, it does not appear to be a systematic review, but you can correct me if I'm wrong.

We included SRs that met the Database of Abstracts of Reviews of Effects (DARE) criteria[8]: 1) reported eligibility criteria, 2) adequate search, 3) data synthesis, 4) risk of bias assessment and/or 5) individual description of included studies. To be included, SRs had to meet at least four of these criteria, the first three of which were mandatory.

Vakili 2020 meets 1, 2, 3 and 5 (narrative and also tabular synthesis at study level in the supplement Table 1) 

• Overall writing: I found that the writing was appropriate for publication, minus a few minor punctuation and writing errors (omitted commas, covid vs. COVID, proper nouns not consistently capitalized). However, I did notice some statements or sections that need clarification:

We only replaced one covid by COVID in the whole manuscript and tried to be consistent with the capitals.

-Table 3, question 5: Clarify statement. I believe there is a word or two missing

We consider that the question is correct “Do mothers transmit SARS-CoV-2 infection to their offspring through breastfeeding?.”

-Lines 166-167: Clarify statement – Table 1 is included and excluded studies? Or is Table 1 included and S4 Table excluded studies?

Your are right. We reworded the sentence:

“Table 1 presents the included studies and S4 Table the list of excluded studies with their exclusion reasons.”

Thanks! Good job and thanks for this very interesting publication.

Thank you for your relevant contributions!

---

## [Editor Report · Decision Letter 1]

17 Jun 2021

COVID-19 and pregnancy: An umbrella review of clinical presentation, vertical transmission, and maternal and perinatal outcomes

PONE-D-21-14273R1

Dear Dr. Ciapponi,

We’re pleased to inform you that your manuscript has been judged scientifically suitable for publication and will be formally accepted for publication once it meets all outstanding technical requirements.

Kind regards,

Linglin Xie

Academic Editor

PLOS ONE
---

## [Editor Report · Acceptance letter]

21 Jun 2021

PONE-D-21-14273R1 

COVID-19 and pregnancy: An umbrella review of clinical presentation, vertical transmission, and maternal and perinatal outcomes 

Dear Dr. Ciapponi:

I'm pleased to inform you that your manuscript has been deemed suitable for publication in PLOS ONE. Congratulations! Your manuscript is now with our production department. 

Kind regards, 

on behalf of

Dr. Linglin Xie 

Academic Editor

PLOS ONE